# Neglect of publication bias compromises meta-analyses of educational research

**Ivan Ropovik** [1,2]*, **Matus Adamkovic** [1,3], **David Greger** [1]

**1** Institute for Research and Development of Education, Faculty of Education, Charles University, Prague, Czechia, **2** Faculty of Education, University of Presov, Presov, Slovakia, **3** Institute of Psychology, Faculty of Arts, University of Presov, Presov, Slovakia

* ivan.ropovik@gmail.com

**Data Availability Statement:** The data underlying the results presented in the study are available from https://osf.io/ephba/.

**Funding:** This work was funded by project PRIMUS/20/HUM/009 and by the Slovak Research

## Abstract

Because negative findings have less chance of getting published, available studies tend to be a biased sample. This leads to an inflation of effect size estimates to an unknown degree. To see how meta-analyses in education account for publication bias, we surveyed all meta-analyses published in the last five years in the Review of Educational Research and Educational Research Review. The results show that meta-analyses usually neglect publication bias adjustment. In the minority of meta-analyses adjusting for bias, mostly non-principled adjustment methods were used, and only rarely were the conclusions based on corrected estimates, rendering the adjustment inconsequential. It is argued that appropriate state-of-the-art adjustment (e.g., selection models) should be attempted by default, yet one needs to take into account the uncertainty inherent in any meta-analytic inference under bias. We conclude by providing practical recommendations on dealing with publication bias.

## Introduction

Because any single study seldom provides conclusive evidence, meta-analyses are seen as the most objective way to settle substantive scientific questions. Meta-analysis is a way to synthesize evidence from a set of studies examining the same underlying effect in a consistent and transparent manner. If such a set of studies comes from a systematic review of the literature, the goal is usually to provide a generalizable inference about a target effect, as it tends to be studied.

In education, field-wide evidence synthesis employing eight hundred meta-analyses leads to the conclusion that almost everything works, with an average effect size of $d$ = .40 [1]. At the same time, pre-registered large scale randomized controlled trials (RCTs) of most promising educational interventions (required by the funders to be published irrespective of the result) yield an average effect size of just $d$ = .06 [2]. Likewise, in psychology, large scale replications carried out by multiple independent labs (published as registered replication reports, i.e., in-principle accepted purely on the merits of methods, prior to data collection) yield overall effects that are three times as small on average as meta-analyses examining the same effect [3]. Thus, there is a discrepancy between the results of meta-analyses (being the reflection of

and Development Agency under Grant no. [APVV-17-0418 and APVV-18-0140]. The funders had no role in study design, data collection and analysis, decision to publish, or preparation of the manuscript.

**Competing interests:** The authors have declared that no competing interests exist.

available primary studies) and rigorous gold-standard studies that are at no risk of being rejected after seeing the results.

As a matter of fact, the full body of published or otherwise available studies represents a biased sample of the conducted (and all conceivable) studies. As non-significant or negative focal findings are less likely to get published, they frequently end up in the file drawer (see [4]), leading to a phenomenon known as publication bias. Direct evidence for publication bias was found by numerous cohort studies examining the publication status of projects that received ethics approval, grant funding, or by tracking study registrations, reports to licensing authorities, and conference abstracts (see [5]). Publication bias is not much different from a situation where a researcher striving to corroborate a theory arbitrarily removes observations contradicting the predictions. Most people would see such a practice of withholding single observations as being at odds with the norms of science. And yet, file-drawering entire studies contingent on the results may still not be fully recognized in the meta-analytic applied practice as a crucial problem for the integrity of scientific knowledge.

Publication bias usually leads to an inflation of estimated average effect sizes to an unknown and possibly substantial degree [6–8] and tilts the evidence in favor of the dominant theory [9]. Even if most of the synthesized studies report significant results and the underlying effect seems large and robust, it may well be zero under substantial publication bias (see [10, 11]). As an example, a meta-analysis of the ego-depletion effect in psychology synthesized 198 effects, of which 85% were statistically significant, yielding an average effect size of $d = .62$ [12]. However, a re-analysis of the data by Carter & McCullough [10] suggested a nil effect, when corrected for publication bias. Subsequent multi-lab ($k = 23$) pre-registered study of the original effect found a non-significant overall $d = .04$ [13].

Simulations also show that under even medium publication bias, naïve meta-analytic estimates unadjusted for publication bias show a substantial upward bias and the false-positive rate approaches 100% with the increasing number of included effects [14, 15]. Instead of factoring out study idiosyncrasies and protecting the consumers of research from single-study biases, an unadjusted meta-analysis (under publication bias) tends to find seemingly precisely estimated, but substantially inflated effects. If there is a publication bias, including more published studies only exacerbates the problem [16].

It is challenging to interpret the results of any single study when there is publication bias (apart from other sources of bias). Nevertheless, there are methods that can, under certain assumptions, help correct for publication bias. A naïve meta-analytic model tends to be a causally misspecified model assuming a frequently untenable publication process–that positive and negative findings have the same chance of getting published. On the other hand, bias correction methods work in various ways to estimate an average effect size under a more plausible implicit data-generating process.

That said, what is the severity of publication bias in education, and how would the conclusions of educational meta-analyses change if we accounted for it? Large-scale meta-studies in education are yet to be carried out, but evidence from other fields suggests that the corrected meta-analytic effect sizes tend to be markedly smaller, and numerous established effects might, in fact, be lacking empirical support. This was found in various fields and domains [10, 11, 17–29]. There is no obvious reason why the domains of educational research should be completely immune to publication bias. As long as the publication of the included results depends on their significance, the current shift towards evidence-informed decision-making may in some instances lead to the adoption of suboptimal or ineffective educational practices, unethical waste of personnel and financial resources, and it may slow down progress in educational research.

In the present meta-study, we took the entire population of recent meta-analyses published in two flagship educational review journals (Review of Educational Research and Educational Research Review) and examined if and how they accounted for publication bias. Arguably, this population of studies should reflect the contemporary state-of-the-art in educational meta-analyses. The present study is therefore not a systematic review. By picking only the most recent meta-analyses from the two highest-impact specialized review journals in the field of education, we aimed to examine the population of state-of-the-art meta-analyses that have the highest potential to steer practices and policies in education. The results thus do not pertain to the entire population of meta-analyses in education. We assume, however, that meta-analyses published earlier or in lower-tier or non-specialized journals do not fare systematically better in terms of dealing with bias.

We studied what proportion of these meta-analyses try to (1) detect bias, (2) correct for it, and (3) explicitly align their substantive conclusions with corrected estimates. We also examined what bias detection and correction methods tend to be used and whether the authors of the meta-analyses tried to search for and include grey literature. In light of the results, we discuss why bias correction should be attempted by default, regardless of the outcomes of bias detection tests. We conclude by laying out some practical recommendations on how to deal with publication bias in evidence synthesis.

## Method

### Search strategy

We reviewed all the empirical papers published from January 2016 to December 2020 in two flagship educational journals specializing in systematic review studies, the Review of Educational Research and Educational Research Review. The only inclusion criterion was the following: the paper had to include a *meta-analysis* providing a quantitative synthesis of *primary studies*. No restrictions on the type of synthesized effects were applied. We excluded meta-research studies on methodological aspects or umbrella reviews (reviews of meta-analyses).

First, we screened all the papers published in regular issues in these two journals within the given time frame. Next, each paper's title and abstract were screened to establish whether it met the inclusion criteria. If so, or when in doubt, we retrieved and screened the full-text to determine eligibility.

### Coding

For each meta-analysis, we manually screened the method and results sections and coded whether the authors used methods for detection and correction of publication bias and reported the results. That did not include the risk of bias assessment at the level of individual studies (evaluation of possible methodological confounding effects related to randomization, blinding, etc.). Apart from hand-screening, we also searched within each paper for terms "bias*", "file*", "sensitivi*", "unpub*", "peer*", "small-study*", "correct*", "adjust*", "selecti*", and for names of mostly used bias detection and correction methods (given below). We binary-coded which of the following (family of) methods for bias detection and correction did the authors directly report: visual inspection of funnel plot asymmetry (plot being available to the reader; [30]), Egger's test [31], rank correlation test [32], correlation between the effect size and $N$, fail-safe $N$ [4], excess significance test [33] for bias detection; trim & fill [34], regression-based model (PET-PEESE; [35]), 1-parameter selection models (*p*-curve, [36]; *p*-uniform, [37]), and 3-parameter selection model (or any other implementation of multiple-parameter selection models; see [7]) for bias correction. We coded the presence of bias detection and correction when the authors reported the results of at least one of these methods. We did not

regard the moderator analysis examining the difference between published and unpublished studies in the reported effect size as a publication bias detection method (but see [38]). Still, we coded if such an analysis was reported.

By screening the abstract, results, and discussion sections, we also examined whether the authors reported detecting publication bias as indicated by at least one the employed methods (e.g., significant *p*-value for bias detection test, Fail-safe *N* smaller than Orwin's criterion) or drew any explicit conclusions regarding the presence of bias informally (e.g., by visual inspection of the included funnel plot). This was considered for the overall effect or for any sub-analysis in which publication bias was specifically addressed. In case that the authors reported detecting publication bias and calculated an adjusted estimate, we coded whether they explicitly informed any of their substantive conclusions by adjusted estimates, that is, whether the correction was consequential for the interpretation of the synthesized data. This was judged by identifying the main verbal interpretation of the focal effect size estimates and the associated inference in the abstract and discussion and comparing those with the results of reported bias correction. Next, we examined whether the authors made an effort to include grey literature or relied exclusively on peer-reviewed journals. Here, we coded 1 if (a) searching for grey literature was mentioned explicitly, (b) if the search strategy covered at least one database in search strategy that indexes unpublished material (e.g., preprint servers, Google Scholar, or ProQuest Dissertations), or (c) if the authors requested unpublished studies from professional LIST-SERVs. We coded 0 only if the authors explicitly stated that unpublished status was an exclusion criterion. Lastly, for each meta-analytic model (excluding meta-regressions) reported in each of the included meta-analyses, we coded the number of included effects meta-analytic models and computed the median *k* for each meta-analysis. For coding the number of included effects *k*, we used tables, forest plots, and searched for the information in the text (in that order). If the number of effects were not reported directly, we tried to estimate *k* based on degrees of freedom for the *Q* test. When it was not clear whether, e.g., the authors used hierarchical models, or averaged the dependent effect sizes, we applied the benefit of the doubt principle and coded the larger number.

Both screening of the papers and data coding were done independently by the first two authors. The mean inter-rater reliability (estimated by Cohen's *κ*, accounting for chance agreement) for the coded variables was .93 (*SD* = .10), ranging from 1 to .74 (lowest for the inclusion of grey literature, representing an agreement of 91%). All the disagreements were subsequently discussed and resolved by consensus prior to the data analysis. Our data are fully open, so all coding decisions can freely be checked. Given the nature of the present survey, we relied solely on the information provided in the papers, without soliciting further clarifications by the authors of the reviewed papers.

The coding scheme, data, and analytic R script for this study are available at osf.io/ephba. We report all the data that we collected in this study. The protocol was not pre-registered, as the present meta-study is not a systematic review and it tested no a priori hypotheses.

## Results

From January 2016 to December 2020, a total of 292 papers were published in the two target journals. From this finite population, we identified 110 eligible meta-analyses (*n* = 54 for Review of Educational Research and *n* = 56 for Educational Research Review). The median number of included effects in the meta-analytic models was *k* = 15 effects (*M* = 36.82, *SD* = 66.23). The plotted results can be seen in Fig 1, and they can briefly be summarized as follows.

First, the survey showed that 18% (*n* = 20) of the meta-analyses did not report addressing publication bias at all, that is, not reporting the use of any bias detection or correction method.

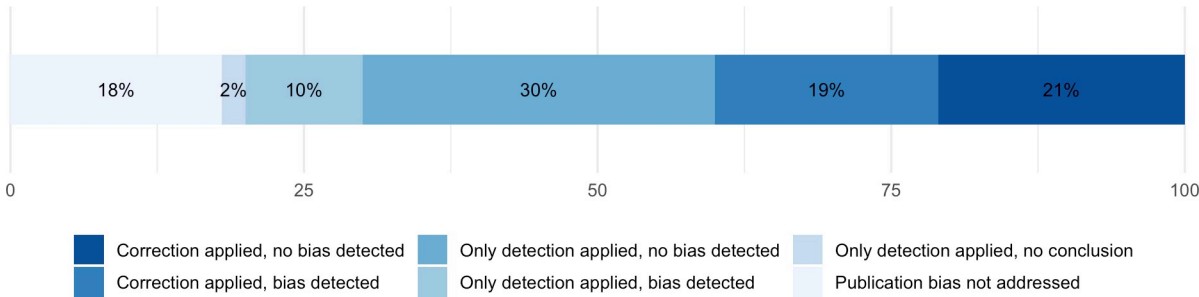

**Fig 1. Treatment of publication bias by recent meta-analyses in education.** Categories are ordered from most favorable (left, top) to least favorable practice/situation (right, bottom).

Less than a half (42%; $n = 46$ in total), tested for the presence of bias but did not employ any adjustment method. The rest (40%, $n = 44$ in total) of the sampled meta-analyses did both, try to detect and adjust for bias.

Second, out of meta-analyses that reported testing for and detecting a significant amount of bias (29%, $n = 32$ in total), 66% tried to adjust for it. Only 9% of the meta-analyses, which corrected for bias (4%, $n = 4$ overall), however, based their substantive interpretation on the adjusted estimates. Half of the meta-analyses (51%, $n = 56$) explicitly concluded that they did not detect publication bias in the first place (based on the use of at least one detection or correction technique). Forty percent of those meta-analyses ($n = 23$) employed publication bias correction method (mostly trim & fill) regardless–but the majority using it, in fact, rather for exploratory purposes of bias detection. Two meta-analyses did not draw any explicit inferences or conclusions regarding the presence of bias despite testing for it (2%). Lastly, most of the meta-analyses (77%, $n = 85$) searched for grey literature in some manner.

In Fig 2, we plotted the frequencies with which bias detection and correction techniques were used in our sample of meta-analyses (with all meta-analyses as the frame of reference). While 18% ($n = 20$) of the surveyed meta-analyses did not use any bias detection or correction method, 28% ($n = 31$) used one such method and 54% ($n = 59$) used more than one method. The three most often employed bias detection techniques were the visual inspection of the

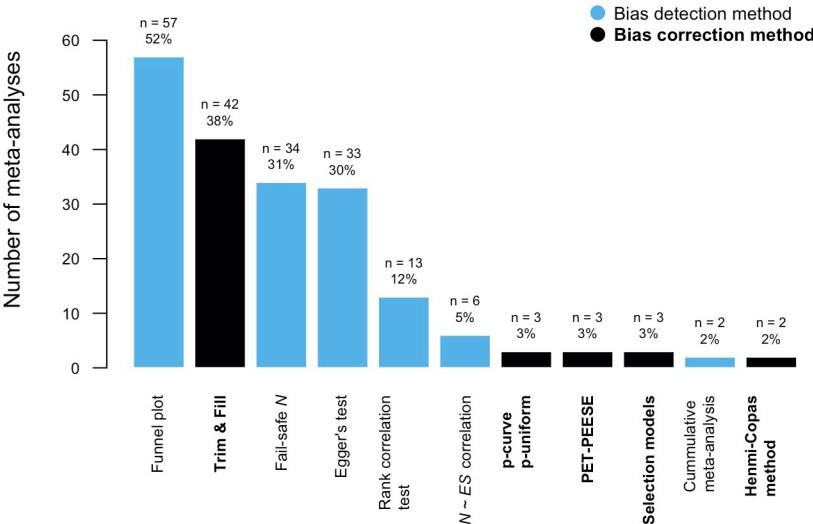

**Fig 2. The use of bias detection and correction methods.**

funnel plot, Fail-safe $N$, and Egger's test, respectively. Twenty-five percent ($n = 27$) compared the magnitude of overall effect sizes between published and unpublished studies. Regarding the use of bias correction methods, 38% ($n = 42$) of meta-analyses reported using the trim & fill. Within the reference class of meta-analyses that set out to correct for publication bias, every single meta-analysis invariably employed the trim & fill method, while 86% used no other correction method but trim and fill. Less than one in ten surveyed meta-analyses that adjusted for the detected bias explicitly aligned their headline quantitative interpretations in the abstract or discussion sections with the results of bias-correction methods.

## Discussion

The present meta-study examined how publication bias was dealt with in the population of meta-analyses reflecting the contemporary state-of-the-art in the field of education. The data show that a large majority of educational meta-analyses addressed the issue of publication bias by employing some of the bias detection methods. Overall, a third of meta-analyses employing bias detection reported detecting publication bias. Out of those meta-analyses, about two in five did not use any correction at all. In all the reviewed meta-analyses, publication bias adjustment was absent in the majority of the meta-analyses. If carried out, it was invariably done using the statistically non-principled trim & fill method. More importantly, it usually lacked any tangible consequences for the headline interpretation of the magnitude of the underlying effect.

The results suggest that even in the population of high-profile educational meta-analyses, the adoption of advances in publication bias adjustment has been rather slow so far. Given that the insights from such adjustments were also rarely factored into the substantive interpretations, the actual evidence backing the conclusions of educational meta-analyses may not be as robust as assumed. Correction efforts were frequently (48%), associated with the detection of bias. Here, 37% of these meta-analyses (31% overall) still employed the widely criticized Fail-safe $N$ test, which result is, in fact, not a function of publication bias and is well known to rest on untenable assumptions, usually leading to fallacious inferences (see [39, 40]). In none of the application did Fail-safe $N$ suggest presence of publication bias.

Although it is difficult to establish any temporal precedence from reported information alone, at least some meta-analysts may have chosen whether to perform bias correction conditional on testing for bias. However, that practice rests on an implicit assumption that the absence of evidence is the evidence of absence, particularly, that the bias detection tests have zero Type II error rate under all conditions. This assumption is untenable. Publication bias is, in fact, a property of the research process, not individual studies. Any product of that process–whether or not it shows evidence of it–will thus be biased [41]. Because sampling a seemingly unbiased set of studies does not make the underlying process unbiased, one should make effort to correct for bias by default. The decision to correct for bias should not be contingent upon the outcome of bias detection. The median number of included effects in meta-analytic models was only $k = 15$. If the number of studies, and thus the power for detecting other than a pronounced pattern of bias is low [14, 42], a null outcome from a bias detection test may provide a false sense of confidence. Even searching for grey literature or increasing the number of included studies cannot solve the problem if the authors' decision to write up the study is being affected by the results [43].

If the volume of evidence is substantive enough for a synthesis to be informative, there is no reason not to attempt to adjust for bias. Here, the results show that only 40% of all meta-analyses reported bias-adjusted estimates. A large majority of these meta-analyses employed only the trim & fill method for bias correction. Under publication bias, this method is, however,

known to suffer from a consistent upward bias and a high false-positive rate similar to that of a naïve meta-analytic estimate [14, 15, 37]. Although better-performing techniques such as the selection models (see [7]) have been around for many years now, only 3% of reviewed meta-analyses used them. Of course, it does not follow that most of the unadjusted meta-analytic estimates are substantially distorted to the degree that their substantive conclusions do not hold. The absence of more rigorous bias correction, however, compromises the integrity of the inferences drawn from educational meta-analyses. Whether the findings of those meta-analyses hold after applying state-of-the-art bias correction methods should, therefore, be of concern.

That said, can the bias correction methods alone solve publication bias and recover the "true" magnitude of a studied effect? Unless we map the system of causal processes behind publication bias, the answer will likely be no. None of the current bias correction methods works sufficiently well under all usual conditions (especially with large heterogeneity or non-normal distribution of true effects). Even the state-of-the-art correction methods may fail to recover the underlying effect [3]. Currently, none of the widely used correction techniques can simultaneously accommodate selection for significance at the level of focal results (publication bias) as well as at the level of analytic pathways leading to those results (the issue of multiple potential comparisons, [44]). In this respect, Friese and Frankenbach have shown that the effect of publication bias gets amplified if some of the null or negative results are transformed into positive results–especially under moderate publication bias and small true effects [45]. Lastly, there is considerable inherent uncertainty associated with the adjusted effect size estimates [14, 15, 37, 46], also leading to lower statistical power for detecting more subtle effect sizes.

Apart from the difficulty to recover the true underlying effect size, the meaning of an average effect cannot be detached from research designs and operationalizations of the included studies. These are, in fact, a non-random sample from a population of all relevant studies that could have been conducted [47]. Meta-analytic estimates are thus not comparable if the effect-size-relevant design features are not uniformly distributed or controlled for (see [48]). All that makes it difficult to assess (1) whether the results of a meta-analysis are robust to bias, (2) to what universe of studies do the results generalize if that is the goal, and (3) what is the theoretical relevance of the meta-analytic average effect sizes (see [47, 48]). Consumers of educational research and meta-analysts should therefore primarily downgrade their expectations concerning the conclusiveness of meta-analyses and not place too much confidence in the meaning and precision of the estimated overall effects.

Large-scale evidence synthesis followed by the ranking of educational interventions is gaining a lot of traction via initiatives like the Teaching and Learning Toolkit in the UK, or What Works Clearinghouse in the US, driving the decision-making of schools and entire educational systems. Understandably, education practitioners, schools, and policymakers demand simple and actionable answers. Refraining from giving such answers when the evidence is not convincing should, however, not be seen as a failure. The imperfection of bias correction methods and the state of the available educational literature frequently do not warrant much beyond examining the gaps in the available evidence, inferring the sources of heterogeneity, and tentatively stating whether the effect (in the way it tends to be studied, on average) is likely to be rather relatively biggish, smallish, negligible, or whether the evidence is inconclusive. Syntheses of educational research may be better off shifting at least some of the focus from trying to recover underlying "true" effect sizes towards a more critical appraisal of the quality and biases of available evidence.

That said, there are things that can be done in practice to tackle publication bias in evidence syntheses. Below, we try to provide some practical recommendations. For a more general set

of recommendations on how to conduct a rigorous meta-analysis, see a comprehensive guidance paper by Pigott & Polanin [49].

1. *Correct for bias. It is far from perfect, but still better than no correction at all.* There are few realistic conditions under which it is preferable to base substantive conclusions solely on naïve, uncorrected meta-analytic estimates. These include (1) conducting a prospective or mini meta-analysis and synthesizing (2) a set of studies conducted as registered reports or (3) "non-headline" effects, that is, effects independent from the focal effect of the paper [50]. Also, in some analytical contexts, the goal may be just to summarize the results of a finite set of studies without attempting to draw generalizable inferences about unobserved studies (using fixed-effects meta-analytic models). The random-effects model can–as any other hierarchical model–then be employed to regularize the estimates. If, on the other hand, the aim is to draw inferences about some underlying population of studies, the random-effects meta-analytic models impose the assumption that all the conducted empirical studies are written up and published *regardless of the outcome*. This assumption is untenable in most fields of social or behavioral science. All bias correction methods also make assumptions that may be problematic in certain contexts and none of them recovers the underlying effect size in a sufficiently reliable way. However, given that the publication bias is ubiquitous in syntheses that include studies' focal effects, and its severity is almost always unknown, formally adjusting for publication bias tends to be less wrong than not doing so.

2. *Try to correct for bias even if none was detected.* Detection of all types of bias has important diagnostic value in appraising the quality of evidence. Informal assessment (eyeballing the funnel plot; see [51, 52]) or testing for the presence of publication bias should, however, not guide the decision whether to adjust or not. Publication bias detection tests have been shown to lack statistical power in many settings (see [42]) and in the case of publication bias, they address a hypothesis that is known to be usually false [41]. Authors and editors alike are known to disfavor negative results [53, 54]. These then represent a tiny fraction of published evidence in social sciences [55]. It is simply not the case that all studies (and effects within those) get published regardless of their results. Such selection always leads to bias. Correction for publication bias should thus practically always be an integral part of a meta-analytic workflow.

That said, bias correction usually requires substantively larger number of included effects to provide an informative estimate. If the number of included independent effects is too small (may also be due to clustering of effects), no proper adjustment may be viable and publication bias needs to be tacked by simpler sensitivity analyses (e.g., [56]). Put simply, sound evidence synthesis of a given research program requires a fair amount of cumulative empirical evidence. Sometimes, the only sound conclusion is that the given research program is not yet ripe for an informative synthesis.

3. *Employ state-of-the-art bias correction methods.* Instead of relying on ad hoc methods with largely unknown statistical properties, it is advisable to base meta-analytic inferences on statistically principled correction methods, like the multiple-parameter selection models. The advantage over many other bias correction methods is that this highly flexible family of models attempts to model the functional form of the biasing selection process directly [see 7].

On the other hand, different implementations of selection models impose different assumptions and have different shortcomings [7]. One of the most general solutions to such an issue is then to average over various models to make bias correction more robust to model misspecification [57, 58]. For selection models, this is now easily done in *R* or in statistical package *JASP* [57]. For a different Bayesian implementation of selection models that allows to model not only publication bias but also p-hacking [59]. In situations where the number of included effects is too small, selection models can even be used as a sensitivity analysis. In the Vevea &

Woods approach [60], one can fit a series of fine-grained step function models where the selection weights are not estimated but defined a prori. In doing so, the model gives up the ability to provide a single best guess of unbiased effect size. On the other hand, it makes it possible to examine the results assuming different forms and severity of bias (e.g., moderate, severe, and extreme selection) [7, 60].

There are also several other approaches to publication bias correction [see 15] and new ones are continually being developed. In many practical applications, it may be sensible to supplement selection models with different approaches. By applying distinct methods, one can effectively explore how much variable are the effect size estimates under different assumptions about the selection process. However, interpreting the degree of concord between different methods assumes that the employed methods tend to perform well in the given analytic scenario. Correction methods can, in fact, disagree with each other not just due to different assumptions but also due to distinct biases [11].

It is therefore important that the methods used for bias correction perform reasonably well in the given specific context. Most correction methods, including selection models, also assume independent effect sizes and are thus not intended for use in effect sets having a multilevel structure. The meta-analyst then needs to be aware that if dependencies are not properly modeled (see [61]) or handled otherwise (e.g., using a permutation procedure, see [62]), inflated type I error rates are to be expected. In general, the choice of bias correction methods requires either making assumptions or having knowledge about a given research program, and should be preceded by a method performance check (see, e.g., http://shinyapps.org/apps/metaExplorer/, an online Shiny app; [14]). We recommend that, at least, when several adequate correction techniques indicate a sharp difference from the crude meta-analytic estimate, such estimate should lose much of its interpretational relevance, and the substantive inferences need to be informed primarily by the results of bias adjustments.

4. *Consider other small-study effects.* There is no substitute for scientific reasoning. Likewise, one always needs to be careful also with declaring the impact of publication bias. In fact, a pattern consistent with publication bias can also arise due to other, entirely legitimate reasons, the so-called small-study effects. Apart from the possibility of genuine random variation, these include: (1) hard-to-reach populations may have a more sensitive response to intervention, and even if not, they may likely show larger effect size because of restricted within-group variances; (2) interventions delivering stronger effects might not be scalable to larger samples–smaller studies may be able to create more favorable conditions; (3) researchers may have a very precise a priori intuition about the magnitude of the true effect and power their designs accordingly; (4) there may be a decline effect where the extreme (overestimated) effects get selected for more extensive study, followed by the regression to the mean; or generally, (5) there may be some other systematic methodological differences between smaller and larger studies that are associated with larger raw effect size or smaller sampling variance [49, 63–65]. It is, therefore, a good practice to check whether the observed bias cannot be explained away by controlling for (or blocking) the variation in methodological aspects of the studies or sample characteristics.

5. *Try to recover and include unpublished studies.* On logical grounds, including as much written-up grey literature as possible cannot eliminate the built-in bias, but it helps. Authors should always try to recover available unpublished material. Yet, this is no exception to methodological principles. Namely, meta-analysts should be cautious to minimize the selection bias caused by overrepresenting their own work, their close collaborators, or known prolific authors. It has actually been shown that authors of meta-analyses are twice as often represented in unpublished than in published studies [66]. If the recovered unpublished studies are not representative of the population of conducted unpublished studies, their inclusion may,

therefore, induce additional bias. That is because being written up, these studies might also have been subject to various forms of selective reporting of focal results (as most were probably intended to be published).

6. *Make meta-analyses reproducible*. As there are usually no ethical, privacy, or legal constraints in sharing the data underlying a meta-analysis, this should be done routinely. At least, it should always be evident, which studies and effects were included and what were the coded effect sizes and their confidence intervals (alternatively variances or standard errors) for these studies [49]. That allows for the reproduction of meta-analytic models and the application of most contemporary bias correction methods. Lakens et al. [67] provide practical guidelines on how to make meta-analysis reproducible.

For consumers of meta-analyses, Hilgard et al. [11] offer an instructive and up-to-date example (with freely available *R* code) of how to reevaluate publication bias in a published meta-analysis. This may be useful before building a research program on a given effect.

7. *Gain credibility by pre-registering and following reporting standards*. Meta-analyses enjoy substantial impact, and their methodological integrity should, therefore, be of utmost importance. For instance, compared to education, fields of medicine and epidemiology have far more experience dealing with various biases involved in primary research. Yet, they started to routinely pre-register meta-analyses only relatively recently, when the production of unnecessary, misleading, and conflicted meta-analyses became too common [68]. In fact, every meta-analyst faces plenty of arbitrary decisions and possibilities for undisclosed flexibility. Combined with the mostly retrospective nature of the evidence synthesis, this easily invites bias.

By outlining the search terms, inclusion criteria, coding decisions, analytic plan, or rules of inference in advance, the meta-analyst reduces the room for confirmation bias and error and buys credibility while still having the chance to revise or amend the protocol along the way transparently [49]. Pre-registration (e.g., on the Open Science Framework or a prospective register of systematic reviews like PROSPERO) does not preclude data exploration. It only marks a demarcation line between exploratory and confirmatory research.

Eventually, all published meta-analyses need to follow established standardized reporting guidelines like the Preferred Reporting Items for Systematic Reviews and Meta-Analyses (PRISMA) [69].

8. *Bias correction should matter and be reported adequately*. In educational policymaking, the reliance on quantitative synthesis should be inversely proportional to the expected severity of bias in the given body of research. Unveiling bias may thus help to critically appraise what we know and what is yet to be known.

Conclusions drawn from publication bias correction should be summarized in the abstract. The paper should also inform the reader about the uncertainty inherent in such adjustments. Just like the unadjusted analysis, bias correction cannot, by any means, provide a single definitive estimate when the available literature is a biased sample.

It does not matter whether bias correction is called or regarded as a sensitivity analysis. It is important that reporting of bias adjustment should be done with the same care as the reporting of unadjusted analyses. Without a potential impact on the meta-analytic conclusions, publication bias correction is useless.

Lastly, tackling publication bias in future research directly by striving for more robust and representative primary evidence is superior to any ex-post corrections. There are several possible ways how to mitigate publication bias at the collective level. First and foremost, there needs to be more openness, transparency, and rigor in primary research. Nowadays, too much research is being conducted, while too little of it is available in writing. Funders may enforce detailed pre-registration of hypothesis-testing studies in public registries and incentivize rigor over quantity. Better-designed, larger, and more transparent studies that are informative

regardless of their outcome will then be less likely to stay unavailable. Institutional review boards may also require researchers to publish (as an article or preprint) or at least write up the results of past research before approving a new study. Educational journals should not prohibit preprints and judge manuscripts on the merits of the methods, not the results. Lastly, the wide adoption of registered reports for confirmatory research and support for open science practices in general [70] are an essential part of the solution.

There is no question that educational policies should be informed by empirical evidence. As the effectiveness of education has wide-ranging societal impacts, changes to educational policies should be backed up by a robust synthesis of all available evidence. Without factoring in the systematic missingness of negative results, this evidence may be based on a futile statistical exercise, impeding progress in educational theory and practice. Neglecting the adjustment for publication bias in meta-analyses or making it inconsequential may then lead to the adoption of ineffective or harmful educational policies. Admittedly, even the best bias correction methods are still far from perfect. But as long as publication bias is present, these methods tend to provide a more realistic picture about the quality of evidence backing a given effect. Bias correction should be seen as an indispensable middle ground, but the ultimate goal must be the prevention of publication bias and more transparency and rigor in educational science in general (see [70]).

## Author Contributions

**Conceptualization:** Ivan Ropovik, Matus Adamkovic.

**Data curation:** Ivan Ropovik, Matus Adamkovic.

**Formal analysis:** Ivan Ropovik.

**Funding acquisition:** Ivan Ropovik, Matus Adamkovic, David Greger.

**Investigation:** Ivan Ropovik, Matus Adamkovic.

**Methodology:** Ivan Ropovik.

**Project administration:** Ivan Ropovik.

**Supervision:** Ivan Ropovik, David Greger.

**Visualization:** Ivan Ropovik.

**Writing – review & editing:** Ivan Ropovik, Matus Adamkovic, David Greger.

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
