## [Decision Letter · Decision Letter 0]

4 Nov 2020

PONE-D-20-29585

Neglect of publication bias compromises meta-analyses of educational research

PLOS ONE

Dear Dr. Ropovik,

Thank you for submitting your manuscript to PLOS ONE. After careful consideration, we feel that it has merit but does not fully meet PLOS ONE’s publication criteria as it currently stands. Therefore, we invite you to submit a revised version of the manuscript that addresses the points raised during the review process.

We look forward to receiving your revised manuscript.

Kind regards,

Vance Berger

Academic Editor

PLOS ONE

Journal Requirements:

2.

1) Please include additional information about the rationale for choosing these specific journals and whether they're representative of a larger number of publications.

2) If you intend to provide a current overview of publication and reporting practices at these journals, please include relevant publications from the last year in your study.

3.Thank you for stating the following in the Author Note Section of your manuscript:

[This work was funded by the Czech Science Foundation under grant no.

21 [P402/12/G130], by the Slovak Research and Development Agency under Grant no. [APVV-

22 17-0418 and APVV-18-0140] and by project PRIMUS/20/HUM/009.]

 [The funders had no role in study design, data collection and analysis, decision to publish, or preparation of the manuscript.]

Additional Editor Comments (if provided):

I agree with Reviewer #1 that bias detection is quite important. In fact, a compelling case can be made that it is of paramount importance, and that any researchers would be negligent for NOT undertaking such an effort. This is what I argued in my 2005 book regarding selection bias specifically, but the sentiment applies equally fervently to ALL types of bias. So while we hope that you will address ALL reviewer comments, please pay special attention to this one. Thank you.

Reviewers' comments:

Reviewer's Responses to Questions

**Comments to the Author**

1. Is the manuscript technically sound, and do the data support the conclusions?

Reviewer #1: Yes

Reviewer #2: Yes

2. Has the statistical analysis been performed appropriately and rigorously? 

Reviewer #1: Yes

Reviewer #2: Yes

3. Have the authors made all data underlying the findings in their manuscript fully available?

Reviewer #1: Yes

Reviewer #2: Yes

4. Is the manuscript presented in an intelligible fashion and written in standard English?

Reviewer #1: Yes

Reviewer #2: Yes

5. Review Comments to the Author

Reviewer #1: This study investigates how publication bias was assessed in educational meta-analyses. It is generally well written, and the code and data are publicly available. Although I disagree with some viewpoints by the authors, I think this study presents some novel results to guide future educational meta-analyses. My detailed comments are as follows.

Page 2, lines 38-41 in the abstract (and several places in the discussion). The authors dislike the trim and fill method, and treat selection models as state-of-the-art methods. I understand that the trim and fill methods have been shown to have limitations in several settings, but I don’t think it is “outdated.” It is intuitive and can be easily implemented by many software programs; it is also currently widely used as shown in this manuscript’s results. The original authors of the trim and fill method suggested it as a way of sensitivity analyses, which could effectively inform whether the meta-result is sensitive to potential publication bias.

On the other hand, the authors recommend the use of selection models. Although I generally agree with this, like the trim and fill method, they also depend on specific assumptions, and the bias-corrected results may not be reliable if the assumptions are seriously violated in a meta-analysis. Also, many selection models exist; the question is which one should be trusted if the authors want to rely on the bias-corrected result.

Page 7, line 151. The authors include the p-curve as a type of selection methods; not sure it is within the list of recommended methods. As far as I know, some people are against this method. See a relevant paper: https://doi.org/10.1371/journal.pone.0149144.

Page 8, lines 171-174. I don’t think the authors should presume that meta-analysts searched for gray literature by default. In medical meta-analyses, researchers usually conduct literature search in PubMed etc. for peer-reviewed studies, but not in Google Scholar, arXiv, or medRxiv, primarily due to concerns about the quality of manuscripts without peer reviews.

Page 14, line 340. I strongly disagree with “Bias detection is pointless, always try to correct for publication bias.” Bias detection indicates whether the meta-result is potentially subject to bias or not; it is of course meaningful because decision makers can use it to assess the reliability of the synthesized evidence. It also gives the direction of the potential bias. If the bias direction is toward the null, decision makers could rate down the evidence; if all the meta-result is nearly unchanged in all sensitivity analyses, it could be treated as reliable evidence.

Moreover, it is not feasible to “always try to correct for publication bias” in many meta-analyses. For example, in medical meta-analyses, the number of included studies is usually quite small, say less than 10 or even 5 studies. In such cases, it is a big statistical change to estimate the complicated selection models; some selection models could even have much more parameters to be estimated than the available studies. In addition, as I mentioned earlier, all selection models depend on specific assumptions, while the true publication bias scenarios vary greatly across meta-analyses. It is also impossible to strictly check those assumptions. There are many selection models, and they likely produce very different bias-corrected results for the same meta-analysis; which model should be trusted? In my view, all statistical methods (including the trim and fill method and the selection methods) may serve as sensitivity analyses; the bias-corrected results should be interpreted on a case-by-case basis by evaluating if the assumptions made in the statistical methods are reasonable for a given meta-analysis.

In the results, it would be helpful to report the information about the number of studies included in the 175 educational meta-analyses. The number of studies is an important factor that affects the statistical powers of statistical tests for publication bias and the uncertainties of the bias-corrected results.

Minor comments

Page 6, line 131. Both (1) and (2), or (1) or (2)? I thought that a meta-analysis must present a quantitative synthesis of primary studies; otherwise, it could be only called a systematic review.

Page 6, line 135. The “;” should be “:”.

Page 6, line 137, “meets” -> “met”.

Page 8, lines 189-190, delete “Data for included meta-analyses are available at osf.io/ephba”; it is repeated (lines 183-184).

Page 8, line 191, “.” instead of “:” after “follows”.

Page 17, line 411, should the “.” before “Cautiously” removed?

Page 19, line 450, what does “only labels it as such” mean?

Reviewer #2: The study presents a systematic review of meta-analysis in select journals that are considered flagship (the objective reason for this selection is not stated). Therefore the authors should not use the meta-analysis/study as the study design throughout the manuscript. A potential bias was introduced by not including all available literature on the subject and this should be stated as the limitation of the study. Furthermore, the Journal requires reports of systematic reviews and meta-analyses to include a completed PRISMA (Preferred Reporting Items for Systematic Reviews and Meta-Analyses) checklist and flow diagram to accompany the main text. Authors must also state in their “Methods” section whether a protocol exists for their systematic review, and if so, provide a copy of the protocol as supporting information.

Please include the title Introduction as required by the Journal.

Please define abbreviations upon first appearance in the text (e.g. pg. 3 randomized control trials RCTs).

Consider shortening the text on pg.5. It is not imperative to state every field the publication bias has been studied. A simple citation would suffice [10,11,17-29].

Please rewrite the second sentence in the Coding paragraph of the Methods section. I believe a part of the sentence is missing.

Figure 1. The squares visualization is not informative enough to get a sense of proportions - bar plots would be better. This article could be informative: Sperandei S. The pits and falls of graphical presentation. Biochem Med (Zagreb). 2014;24:311-320

Figure 2. The explanation on y-axis is missing. The figure should be self-explanatory.

On pg. 11, 1st paragraph - "The question then is... Probably not.." Actually, this is not the question of this study, so please omit these two sentences.

In the discussion the authors focus more on giving the recommendations how to deal with the publication bias in general which is not reflected in the title of the study. The results of the study seem merely an afterthought, particularly concearning that none of the included meta-analysis adjusted their conclusions based on the detected bias. The authors should discuss their results and how this potentially affects the field of educational research. Avoid repeating the results (numerically) and instead focus on the interpretation of the data.

The recommendations (elaborations) could be concised for better impact.

The conclusion is supported by the results in the text and the abstract, however the authors again mostly focus on general problem of publication bias. Considering the study focused on educational research so should be the conclusion.

I am not an expert in statistics therefore I cannot review the authors claims about statistics in meta-anaylsis.

The manuscript is written in standard English, however I am not a native English speaker. Perhaps some sentences could be shortened for easier reading flow, but I am leaving this to authors' discretion.

6. PLOS authors have the option to publish the peer review history of their article (what does this mean?). If published, this will include your full peer review and any attached files.

Reviewer #1: No

Reviewer #2: No

---

## [Author Response · Author response to Decision Letter 0]

4 May 2021

Please see the Response letter. Thank you.

---

## [Decision Letter · Decision Letter 1]

17 May 2021

Neglect of publication bias compromises meta-analyses of educational research

PONE-D-20-29585R1

Dear Dr. Ropovik,

We’re pleased to inform you that your manuscript has been judged scientifically suitable for publication and will be formally accepted for publication once it meets all outstanding technical requirements.

Kind regards,

Vance Berger

Academic Editor

PLOS ONE

Additional Editor Comments (optional):

Reviewers' comments:

Reviewer's Responses to Questions

**Comments to the Author**

1. If the authors have adequately addressed your comments raised in a previous round of review and you feel that this manuscript is now acceptable for publication, you may indicate that here to bypass the “Comments to the Author” section, enter your conflict of interest statement in the “Confidential to Editor” section, and submit your "Accept" recommendation.

Reviewer #2: All comments have been addressed

2. Is the manuscript technically sound, and do the data support the conclusions?

Reviewer #2: Yes

3. Has the statistical analysis been performed appropriately and rigorously? 

Reviewer #2: Yes

4. Have the authors made all data underlying the findings in their manuscript fully available?

Reviewer #2: Yes

5. Is the manuscript presented in an intelligible fashion and written in standard English?

Reviewer #2: Yes

6. Review Comments to the Author

Reviewer #2: (No Response)

7. PLOS authors have the option to publish the peer review history of their article (what does this mean?). If published, this will include your full peer review and any attached files.

Reviewer #2: No

---

## [Editor Report · Acceptance letter]

24 May 2021

PONE-D-20-29585R1 

Neglect of publication bias compromises meta-analyses of educational research 

Dear Dr. Ropovik:

I'm pleased to inform you that your manuscript has been deemed suitable for publication in PLOS ONE. Congratulations! Your manuscript is now with our production department. 

Kind regards, 

on behalf of

Dr. Vance Berger 

Academic Editor

PLOS ONE